# The Frequency of Meal-Replacement Products Drinking and All-Cause, CVD, and Cancer Mortality

**DOI:** 10.3390/nu16213770

**Published:** 2024-11-02

**Authors:** Yuxuan Zhao, Aolin Li, Haiming Yang, Meng Xiao, Mingyu Song, Zilun Shao, Rong Jiao, Yuanjie Pang, Wenjing Gao, Tao Huang, Jun Lv, Liming Li, Canqing Yu, Dianjianyi Sun

**Affiliations:** 1Department of Epidemiology and Biostatistics, School of Public Health, Peking University, Beijing 100191, China; 2411110190@pku.edu.cn (Y.Z.);; 2Peking University Center for Public Health and Epidemic Preparedness & Response, Beijing 100191, China; 3Key Laboratory of Epidemiology of Major Diseases, Peking University, Ministry of Education, Beijing 100191, China

**Keywords:** meal-replacement, mortality, CVD, cancer

## Abstract

Objectives: Our study aimed to assess the associations between meal-replacement (MR) drinking and risks of all-cause, cardiovascular and cerebrovascular disease (CVD), and cancer mortality. Methods: The study was based on 6770 adults aged 20 years or older from the National Health and Nutrition Examination (NHANES) 2003–2006 with linked mortality data from the National Death Index for linked mortality records (until 31 December 2019). Respondents were categorized into four groups according to the frequency of MR drinking: ≤1 time per month (seldom), 2–3 times per month (monthly), 1–6 times per week (weekly), and ≥1 time per day (daily). The adjusted hazard ratios (aHRs) of MR drinking with all-cause, CVD, and cancer mortality were estimated by Cox proportional hazards regression models. Likelihood ratio tests were used to find potential interactions of MR drinking with age, sex, and BMI. Results: During a median follow-up of 14.4 years, a total of 1668 death events were recorded among the study population. Compared to respondents who seldom drank MR, daily and weekly drinkers had greater risks of all-cause mortality (aHRs and 95% confidence intervals [CI]: 1.52 [1.17–1.97] for daily; 1.54 [1.24–1.91] for weekly). Stratified analyses indicated that the effects of MR on all-cause mortality were different between females and males and were more substantial among females (*P* for interaction: 0.003; daily female drinkers vs. daily male drinkers: 2.01 [1.40–2.90] vs. 1.24 [0.85–1.81]; weekly female drinkers vs. weekly male drinkers: 1.68 [1.26–2.24] vs. 1.36 [0.97–1.91]). Conclusions: Daily and weekly MR drinking might increase the risk of all-cause mortality.

## 1. Introduction

Meal-replacement (MR) products are processed products containing essential nutrients and meager calories that can be substituted for one or two meals in one day [1]. Evidence has proven its effectiveness in weight control, especially for patients with diseases in need of food restriction (such as type 2 diabetes), and that they are safe to use under medical supervision or specialized guidance for patients suffering from severe diseases and being unable to acquire adequate nutrition from normal food [2,3,4,5].

While many studies have reported their safety, reliability, and efficacy in randomized trials [6], there are also concerns regarding the long-term health effects of MR products. MR products are usually rich in protein and low in lipids, which could induce deficiencies in micronutrients [7]. It has also been reported that users drinking MR products for weight-loss purposes might suffer substantial weight regain in the long-term and subsequently suffer from health problems [8]. Although there are studies reporting the associations of ultra-processed foods and energy-restricted diets [9,10,11] with elevated risks of mortality, evidence assessing the long-term effects of MR is still insufficient. Moreover, previous studies have mainly been based on clinical trials, indicating that the uses of MR products in these studies were usually under strict supervision, and their target populations were usually subjects with obesity or other mild or severe health problems, restricting the generalizability of these studies to the general population. Therefore, as the market for MR products is expanding and more weight-loss programs are applying these products [12], it is essential and of great importance for public health to investigate the effects of MR products on health outcomes in a more general situation.

To address the evidence gap, our study presented a prospective analysis to assess the associations of MR drinking with all-cause, CVD (including heart and cerebrovascular diseases), and cancer mortality based on the nationally-representative dietary and mortality data from the National Health and Nutrition Examination Survey (NHANES) 2003–2006. We also tried to evaluate the potential modification effects of other factors.

## 2. Materials and Methods

### 2.1. Study Population

The NHANES is a program of cross-sectional studies designed to evaluate the health and nutritional status of adults and children in the United States. The program is conducted in 2-year cycles in counties across the U.S. using complex, multistage, probability sampling designs to generate nationally representative estimates, with nearly 10,000 people examined in each cycle. The survey content includes interviews on sociodemographic characteristics, diet, health conditions, and medical and physiological measurements as well as laboratory tests. All participants provided written informed consent, and the protocols were approved by the research ethics boards of the National Center for Health Statistics. More details of the NHANES are available on the website of the Centers for Disease Control and Prevention [13].

Analyses were performed using data from respondents from the NHANES cycles 2003–2004 and 2005–2006 (*n* = 20,470). After excluding respondents without food frequency data (*n* = 7985), respondents aged younger than 20 years (*n* = 5587), respondents with ineligible mortality data (*n* = 7), and those with undefined MR drinking records (*n* = 121), 6770 respondents remained for association analyses (Appendix A).

### 2.2. The Frequency of MR Drinking

The NHANES study included a 24-h dietary recall interview. A second recall interview via telephone 3 to 10 days after the first collection has been added since 2002 to ensure accuracy. In the first dietary interview, participants were interviewed in a private room in the mobile examination center (MEC) by trained assistants, with measuring guide tools to estimate the volume and dimensions of food items.

An NHANES food frequency questionnaire that estimated the frequency of MR drinking was included in NHANES 2003–2006 to collect additional data on food frequency during the 12 months prior to the interview. The NHANES food frequency questionnaire was offered to respondents after completing the MEC dietary interview and was returned to the home office. The question on MR was as follows: “How often did you drink meal replacement, energy, or high-protein beverages such as Instant Breakfast, Ensure, Slimfast, Sustacal or others?”. The frequency of MR drinking was divided into four categories: seldom (never drink MR or 1 time per month or less), monthly (2–3 times per month), weekly (1–6 times per week), and daily (>=1 time per day). The “seldom” category was chosen as the reference group.

### 2.3. Mortality Data

The mortality data were provided by the Centers for Disease Control and Prevention through linking the NHANES respondent sequence number. Data can be accessed at https://www.cdc.gov/nchs/data-linkage/mortality-public.htm (accessed on 8 May 2023).The mortality information included the leading cause of death, follow-up time, and 10 underlying causes of death. In the present study, we used all-cause, CVD (including heart and cerebrovascular diseases), and cancer mortality. In our study, participants with eligible mortality records were retained. The follow-up time was defined as the time from NHANES examination to the occurrence of outcomes or the end of follow-up on 31 December 2019.

### 2.4. Demographic Characteristics, Disease Histories, and Lifestyles

We included several potential confounders including demographic characteristics, disease histories and lifestyles: age, sex (male and female), race (Mexican American, Non-Hispanic White, Non-Hispanic Black, and Others), educational levels (less than high school, high school, and above high school), occupations (seven groups were created according to Bian Liu et al. [14]), family income (less than 25,000 dollars per year, 25,000–75,000 dollars per year, and above 75,000 dollars per year), smoking status (never smokers who smoked less than 100 cigarettes during one’s lifetime, former smokers, and current smokers), drinking habits (never drinkers, current drinkers, and participants whose drinking status was not known), and the intensity of physical activity (optimal: >=150 min/week of moderate activity or >=75 min/week of vigorous activity or combined, intermediate: 1–149 min/week of moderate activity or 1–74 min/week of vigorous activity or combined, and poor: no reported moderate or vigorous activity). BMI categories (measured through weight and height by trained staff; <25 kg/m^2^ and >=25 kg/m^2^). Chronic obstructive pulmonary diseases (COPDs), cancer, and cardiovascular and cerebrovascular diseases were defined using self-reported disease conditions, and laboratory test data were also combined to determine hypertension, hyperlipidemia, and diabetes status. To account for the dietary quality of the respondents, we additionally calculated the Healthy Eating Index (HEI) using the algorithm introduced elsewhere [15], which aims to evaluate the diet quality through the intake of several dietary components from 24-dietary interviews according to the Dietary Guidelines for Americans (DGA). We chose the simple scoring algorithm as recommended, and higher scores of HEI reflected increased intakes in total fruits, whole fruits, total vegetables, greens and beans, whole grains, dairy, total protein foods, seafood and plant proteins, and fatty acids. The means of the two 24-h dietary interviews for energy, protein, total sugars, fat, and fiber intake were calculated in each cycle for each respondent. Serum albumin levels were measured from refrigerated serum using a Beckman Synchron LX20.

### 2.5. Statistical Methods

To estimate the overall percentage of people drinking MR products in the U.S. in the two year cycles and adjust for the clustered sampling design, we first conducted a weighted analysis among respondents with reported MR drinking reports and non-missing weight values for the food frequency section.

For our study population, the characteristics of respondents were described using the means (standard deviations) for continuous variables and frequencies (percentages) for categorical variables. Analyses of covariance, and chi-square tests were applied to assess the differences among groups. Least-squared (LS) means of energy intake, total sugars, protein, fat, dietary fiber, and BMI adjusted for age and sex were evaluated by the R package “lsmeans”, and the results were presented on bar charts.

Cox regression models adjusted for different covariates were fitted to examine the associations and trend associations of MR drinking with all-cause mortality and cause-specific mortality, with follow-up time as the underlying time-scale: (1) In model 1, we adjusted for age and sex; and (2) in multivariate adjusted models, we adjusted for age, sex, race, educational level, occupation, family income, BMI, physical activity, smoking status, drinking status, disease histories (cardiovascular diseases, cancer, diabetes, hypertension, COPD, hyperlipidemia), and HEI. For trend associations, we first assigned a score to each frequency level [13] (Appendix A) according to the analytic notes on the website and then calculated the means of the scores for each frequency group. Using these means as quantitative variables to fit the Cox regression models, we obtained *p* values for the trend associations. We also conducted stratified analyses to find potential interactions of MR drinking on all-cause mortality with age, sex, BMI, and disease histories by likelihood ratio tests. Considering the sample size, we did not test the interactions of MR drinking with other factors.

To validate our results, we then conducted several sensitivity analyses. To preserve the sample size, we first used the R package “mice” to conduct multiple imputation for respondents without MR drinking records. Proportional odds models were used and all covariates with Nelson–Aalen estimators of cumulative hazard were included. After imputation, 9503 respondents were retained for prospective analysis. Then, we combined the “weekly” and “daily” groups to increase the sample size. Third, we excluded deaths within the first two years (*n* = 172) to test whether our results were robust. Fourth, when calculating aHRs for CVD and cancer death, we excluded respondents with CVD (*n* = 911) or with cancer (*n* = 672) at the baseline, respectively. We also excluded diabetes patients as well as respondents who were in diabetic diets or other kinds of special diets at the baseline (*n* = 1153) to avoid potential reverse causation, since diabetes is one of the main chronic diseases for which weight and dietary management are needed, and MR has been proven to be effective in weight management among patients with diabetes [16]. Sixth, we kept respondents who had weight loss or weight retain intentions to test whether the results were robust among respondents with specific weight management purposes (*n* = 3082). Seventh, we excluded participants with implausible energy intake (males <=800 kcal/day or >= 4000 kcal/day; females <= 500 kcal/day or >= 3500 kcal/day; *n* = 253) [17]. Eighth, since the serum albumin levels are related to internal nutrition levels, we excluded respondents with abnormal low levels of serum albumin (serum albumin < 35 g/L) or missing serum albumin values (*n* = 636) to exclude potential respondents who were likely to be affected by severe diseases and unable to acquire adequate nutrition. Finally, we excluded respondents who reported being unable to eat often due to problems of teeth, mouth, or dentures to avoid extreme health conditions (*n* = 421).

All of the statistical analyses were conducted using R 4.3.1. A *p* value below 0.05 was considered to be statistically significant.

## 3. Results

### 3.1. Main Results

In our weighted analysis, the overall estimated percentages of people drinking MR products monthly, weekly, and daily were 5.0%, 5.7%, and 2.6%, respectively, for the 2003–2004 cycle and 5.7%, 5.9%, and 3.3%, respectively, for the 2005–2006 cycle (Figure 1). No sex difference in the frequency of MR drinking was observed for each cycle.

In our study population, we observed differences among the four groups in age, ethnic groups, BMI, physical activities, some medication histories (CVD, diabetes, hypertension, and hyperlipidemia), and total sugar intake (Table 1) at the baseline. After adjusting for age and sex, there were no differences in the LS means for BMI or the intakes of total energy, total sugars, protein, fat, and fiber among the four groups after Bonferroni correction (Figure 2; Bonferroni *p* value: 0.05/6 = 0.008).

In total, 1668 death events were recorded in the study population during a median follow-up of 14.4 years: 577 were caused by CVD and 345 by cancer. In the age- and sex-adjusted models, we found significant associations of MR drinking with risks of all-cause and CVD mortality. In the multivariable-adjusted models, the adjusted hazard ratios (aHRs) for respondents drinking MR products monthly, weekly, and daily were 1.26 (95% CI: 1.01–1.58), 1.54 (1.24–1.91), and 1.52 (1.17–1.97) for all-cause mortality, *P*trend < 0.001; 1.11 (0.74–1.65), 1.37 (0.92–2.03), and 1.70 (1.11–2.61) for CVD mortality, *P*trend = 0.008; 1.49 (0.94–2.36), 1.30 (0.79–2.13), and 0.89 (0.43–1.83) for cancer mortality, *P*trend = 0.972 (Table 2). In the stratified analyses, we found significant interactions of MR drinking with sex on all-cause mortality (Table 3, *p* for interaction: 0.003). The effects were stronger among females than males for weekly and daily drinkers. The aHRs and 95%CI for weekly and daily female drinkers were 1.68 (1.26–2.24) and 2.01 (1.40–2.90), respectively. The aHRs and 95%CI for weekly and daily male drinkers were 1.36 (0.97–1.91) and 1.24 (0.85–1.81), respectively.

### 3.2. Sensitivity Analyses

In the sensitivity analyses (Appendix A), we first tried to preserve the sample size by using multiple imputation for respondents with missing MR records. A total of 9503 respondents were retained, and the results remained almost the same. Next, we combined the ‘weekly’ and ‘daily’ groups and found that the combined group was still associated with higher risks of all-cause (HR = 1.53, *p* < 0.001) and CVD mortality (HR = 1.50, *p* = 0.008). When excluding deaths within the first two years, our results remained consistent. To control for the potential reverse causality caused by baseline diseases, we excluded CVD and cancer cases at the baseline, respectively, and found that the association of daily drinkers with CVD mortality turned out to be insignificant (HR = 1.55, *p* = 0.151). We further excluded respondents with diabetes and respondents in diabetic diets or other kinds of special diets at the baseline, and the associations of weekly and daily MR drinking with all-cause mortality remained significant (HR = 1.54, *p* = 0.001 for weekly; HR = 1.37, *p* = 0.048 for daily), but the association of daily drinkers with CVD mortality turned out to be insignificant (HR = 1.32, *p* = 0.320). In respondents who had weight management purposes, the associations of weekly MR drinking with all-cause mortality were significant (HR = 1.50, *p* = 0.020), and the associations of daily MR drinking with all-cause mortality tended to be positive but insignificant (HR = 1.48, *p* = 0.237). After excluding respondents with implausible energy intake, the results were similar to the main results. When we excluded respondents with abnormal low levels of serum albumin (serum albumin < 35 g/L) or missing serum albumin values, the results remained consistent with the main results. The results did not change much after excluding respondents who reported being unable to eat often due to problems of teeth, mouth, or dentures.

## 4. Discussion

According to the nationally representative data, our study illustrated the general situation for MR drinking in the two study cycles and found that daily drinkers were at higher risks of all-cause and CVD mortality after adjusting for potential confounders.

The dietary assessment in NHANES gave us a brief overview of MR drinking in the U.S. in the early 2000s. According to our estimation, approximately 13.3% and 14.9% of people drank MR products more than once per month in 2003–2004 and 2005–2006, respectively, indicating that these products might not have been trendy at that time. However, recently, a higher prevalence of health issues such as obesity has driven the rising demand for these products. According to the global MR market size by Grand View Research, the market size is forecasted to reach USD 25.02 billion by 2025, with a predicted annual growth rate of 6.5% from 2019 to 2025. Furthermore, America occupies the majority of the market share [18]. However, since we did not find records for NHANES food frequency data in the subsequent surveys from 2007 in NHANES, we could not construct the developing trend for MR drinking from the nationally representative data.

There have been little observational studies directly assessing the long-term health effects of MR drinking. Previous conclusions about MR products have mostly been drawn from randomized trials, where the focus has usually been placed on the efficacy of MR products on weight management or their safety for medical use [3,6,16,19,20,21]. One systematic study summarized that incorporating MR into weight loss programs would benefit the users by reducing weight and cardiometabolic risks within a relatively short time of follow-up (longest follow-up time included: ≤52 weeks) [22]. However, another systematic review pointed out that the conclusions are inconsistent regarding long-term weight loss (>1 year) [23]. Furthermore, the follow-up time of these trials were usually inadequate to accumulate sufficient disease outcomes or mortality. The food frequency data from NHANES 2003–2006 and the linked mortality data allowed us to have sufficient follow-up time (median: 14.4 years) to evaluate the associations between MR and mortality during a more extended period and suggest its potential long-term health effects in a more general population. In addition, in our multivariate-adjusted models, we additionally adjusted for the HEI of each respondent and showed that the effects of MR on mortality were independent of dietary quality. We also conducted several sensitivity analyses to validate our results. However, it should be acknowledged that only a limited number of respondents were included in our analysis. Even though we did try to maximize our sample size using multiple imputation in the sensitivity analysis, the prospective results should be validated in larger samples and more studies.

It is worth noting that the associations of MR drinking with mortality might not be caused by abnormal nutrition intakes or imbalanced BMI levels across different MR drinking groups. In the sketchy estimation for intakes of total energy, total sugars, protein, fat, and fiber, we did not observe differences in them across different groups after adjusting for age and sex. The results suggest that MR drinking might not influence nutrient intake nor lead to lower or higher levels of nutrient intakes compared to normal diets under natural circumstances. For BMI levels, people with high BMI are usually considered to be associated with higher risks of metabolic diseases, and thus perhaps at higher risk of mortality [24]. If people with obesity tend to drink more MR products for weight-control purposes, the effects of MR on health outcomes would be explained by the reverse causation caused by BMI levels [25]. However, our results showed that the adjusted means of BMI were approximately the same across different groups, and the possibility for BMI-induced reverse causation could be ruled out. Furthermore, no interactions of MR drinking with BMI were observed, which suggest that MR might have the same impact on mortality in respondents at different BMI levels.

One possible explanation for the effects of MR on health outcomes is the affected bioavailability of components used in MR. As processed food, the natural components used to make MR products would go through multiple processes, which could influence the bioavailability and digestion of nutrients and even influence the gut microbiome [26,27], and consequently influence body health. As one kind of high protein product, several observational studies have proposed that long-term high-protein intake might contribute to the decline in renal function, and finally to increasing the risk of chronic kidney disease (CKD) in individuals with or without preexisting CKD due to intraglomerular pressure caused by high-protein intake [28]. The changes in renal function might perhaps subsequently lead to increased risks of cardiovascular diseases and diverse health conditions [29,30]. On the other hand, since no difference was found in the intakes of energy and nutrients across different MR drinking groups, our study suggests that MR drinkers might acquire energy and nutrient intake from other sources of food besides MR. The food sources could be further clarified in the future because they could be the reason for the indirect effects of MR on mortality [9]. In short, the clear mechanisms of MR on health outcomes are not fully understood, and future investigations are needed. However, our study still suggests that MR should be used at proper frequency.

We also tried to observe interactions of MR drinking with age and sex on all-cause mortality, but we only found significant results for sex. For females, higher risks of all-cause mortality were observed for weekly and daily drinkers than for males, while males drinking MR products monthly were at a higher risk than females. Future work could pay more attention to this aspect so that the health problems caused by MR products could be more targeted for prevention.

Our study may have important implications for public health as well as future investigations into this topic. First of all, previous studies have mostly reported the benefits of MR use, but our study pointed out that the frequent use of MRs could increase the risk of all-cause mortality. Therefore, our study may raise concerns of MR users about the potential long-term health problems caused by frequently drinking MRs, especially for daily drinkers. For future studies, our study indicates that an extended follow-up time would be needed to fully assess the impacts of MRs on health, and our results might be of interest to studies including cohorts who wish to conduct nutrition- or dietary behavior-related surveys.

The main advantages of our study included the NHANES datasets we used, which provided us with a wealth of research resources and adequate follow-up time to accumulate health outcomes as well as enabling us to apply our conclusions to a more general population. However, the limitations should also be considered. First and foremost, the interviews were conducted nearly 20 years ago and were not timely enough to reflect the current MR market, but the products contained in the questionnaire are still on the market. Second, there are only records on the frequency of MR drinking without detailed information about the type, amount, duration, and reasons for drinking MR products. Third, the dietary interviews were self-reported, which may cause recall bias. Fourth, only two specific causes of death were included due to the limited number of cases of other causes, and it is still not clear how MR drinking is associated with mortality from other causes.

## 5. Conclusions

Daily and weekly MR consumption were associated with higher risks of all-cause mortality, and there might be interactions between MR consumption and sex on all-cause mortality. More prospective research is needed in this field.

## Figures and Tables

**Figure 1 nutrients-16-03770-f001:**
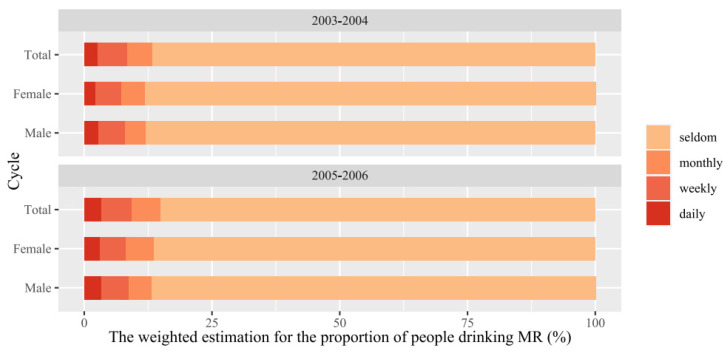
The weighted estimation for the proportion of people drinking MR products in the U.S. in the two year cycle.

**Figure 2 nutrients-16-03770-f002:**
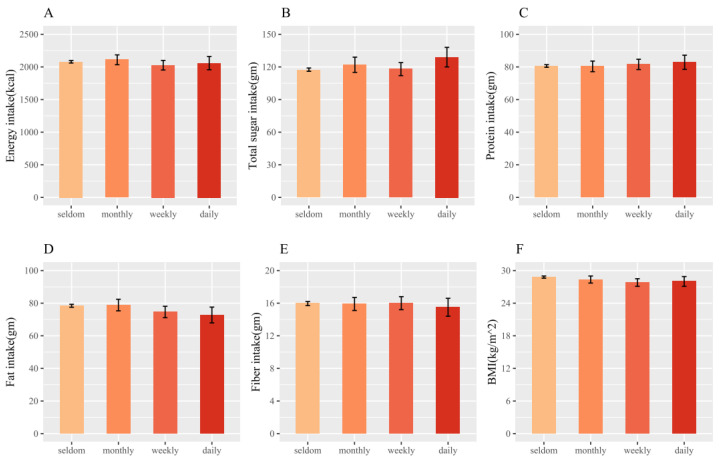
Least-squared (LS) means of intakes of total energy (**A**), sugar (**B**), protein (**C**), fat (**D**), fiber (**E**), and BMI (**F**) by the frequency of MR drinking among 6770 respondents. LS means were adjusted for age and sex. Intakes of energy, total sugar, protein, fat, and fiber were measured by the means of the two day dietary interviews. BMI was measured through weight and height by trained staff. The error bar represents the 95% confidence intervals of the adjusted means. Nutrient intake was calculated from 24-h dietary recall.

**Table 1 nutrients-16-03770-t001:** Baseline characteristics of the study population by the frequency of MR drinking.

	The Frequency of MR Drinking				*P* ^1^
	Seldom	Monthly	Weekly	Daily	
Sample size	5821 (86.0)	365 (5.4)	384 (5.7)	200 (2.9)	
Age (mean (SD))	51.3 (19.0)	48.9 (19.0)	47.5 (19.5)	52.6 (20.6)	<0.001
Female (%)	3113 (53.5)	211 (57.8)	213 (55.5)	100 (50.0)	0.243
BMI (%)					0.003
<25 kg/m^2^	1719 (29.5)	112 (30.7)	140 (36.5)	77 (38.5)	
>=25 kg/m^2^	4018 (69.0)	246 (67.4)	237 (61.7)	117 (58.5)	
Not defined	84 (1.4)	7 (1.9)	7 (1.8)	6 (3.0)	
Physical activity (%)					<0.001
No	2309 (39.7)	124 (34.0)	127 (33.1)	88 (44.0)	
Moderate	1515 (26.0)	87 (23.8)	78 (20.3)	38 (19.0)	
Vigorous	1993 (34.2)	154 (42.2)	179 (46.6)	74 (37.0)	
Not defined	4 (0.1)	0 (0.0)	0 (0.0)	0 (0.0)	
Education (%)					0.514
<High school	1517 (26.1)	92 (25.2)	93 (24.2)	60 (30.0)	
High school	1482 (25.5)	80 (21.9)	94 (24.5)	48 (24.0)	
>High school	2818 (48.4)	192 (52.6)	197 (51.3)	92 (46.0)	
Not defined	4 (0.1)	1 (0.3)	0 (0.0)	0 (0.0)	
Family income (%)					0.308
<25,000 dollars per year	2016 (34.6)	124 (34.0)	149 (38.8)	85 (42.5)	
25,000–75,000 dollars per year	2499 (42.9)	164 (44.9)	156 (40.6)	73 (36.5)	
>75,000 dollars per year	1099 (18.9)	64 (17.5)	63 (16.4)	33 (16.5)	
Not defined	207 (3.6)	13 (3.6)	16 (4.2)	9 (4.5)	
Ethnic groups (%)					0.004
Mexican American	1031 (17.7)	72 (19.7)	87 (22.7)	40 (20.0)	
Non-Hispanic Black	1091 (18.7)	93 (25.5)	84 (21.9)	43 (21.5)	
Non-Hispanic White	3293 (56.6)	178 (48.8)	187 (48.7)	103 (51.5)	
Others	406 (7.0)	22 (6.0)	26 (6.8)	14 (7.0)	
Occupation (%)					0.331
Agriculture	50 (0.9)	1 (0.3)	4 (1.0)	2 (1.0)	
Extractive, construction, and repair occupations	333 (5.7)	16 (4.4)	25 (6.5)	11 (5.5)	
Management	326 (5.6)	21 (5.8)	25 (6.5)	11 (5.5)	
Operators, Fabricators, and Labor	464 (8.0)	34 (9.3)	34 (8.9)	18 (9.0)	
Service	609 (10.5)	46 (12.6)	47 (12.2)	21 (10.5)	
Professional Specialty	534 (9.2)	42 (11.5)	34 (8.9)	8 (4.0)	
Support	766 (13.2)	51 (14.0)	55 (14.3)	21 (10.5)	
Not defined	2739 (47.1)	154 (42.2)	160 (41.7)	108 (54.0)	
History of diseases					
CVD (%)	771 (13.2)	39 (10.7)	44 (11.5)	36 (18.0)	0.004
Cancer (%)	580 (10.0)	27 (7.4)	35 (9.1)	20 (10.0)	0.291
Diabetes (%)	795 (13.7)	37 (10.1)	26 (6.8)	26 (13.0)	0.017
Hypertension (%)	2547 (43.8)	136 (37.3)	136 (35.4)	77 (38.5)	<0.001
COPD (%)	275 (4.7)	13 (3.6)	14 (3.6)	13 (6.5)	0.697
Hyperlipidemia (%)	3567 (61.3)	226 (61.9)	201 (52.3)	111 (55.5)	0.020
Nutrients intake ^2^					
Energy (kcal, mean ± sd)	2053.8 (815.5)	2092.8 (1 004.5)	2044.1 (877.9)	2034.6 (859.3)	0.813
Protein (g, mean ± sd)	79.7 (33.7)	79.5 (40.0)	82.0 (39.1)	82.1 (39.7)	0.474
Total sugars (g, mean ± sd)	116.1 (66.2)	122.0 (74.1)	119.8 (71.2)	127.5 (72.4)	0.035
Fat (g, mean ± sd)	77.5 (37.2)	78.2 (44.2)	75.2 (37.8)	71.9 (37.2)	0.128
Fiber (g, mean ± sd)	15.9 (8.1)	15.7 (8.9)	15.9 (8.6)	15.5 (9.0)	0.917
Healthy Eating Index (mean ± sd) ^2^	48.7 (14.5)	48.6 (14.7)	49.1 (14.2)	50.2 (14.5)	0.449

^1^ *P* values were calculated through the analysis of variance for continuous variables or chi-square tests for categorical variables. ^2^ Nutrient intake and HEI indices were calculated from 24-h dietary recall.

**Table 2 nutrients-16-03770-t002:** Associations of MR drinking with risks of all-cause, CVD, and cancer mortality.

		Number of Events (Incidence Density ^1^)	Age and Sex Adjusted		Multivariate Adjusted ^2^	
			HR(95% CI)	*P*	HR(95% CI)	*P*
All-cause mortality	Seldom	1427 (18.4)	1.00		1.00	
	Monthly	84 (17.6)	1.19 (0.96–1.49)	0.117	1.26 (1.01–1.58)	0.039
	Weekly	91 (18.1)	1.43 (1.16–1.77)	0.001	1.54 (1.24–1.91)	<0.001
	Daily	66 (27.6)	1.68 (1.31–2.14)	<0.001	1.52 (1.17–1.97)	0.002
			*P*_trend_ < 0.001		*P*_trend_ < 0.001	
CVD mortality	Seldom	499 (6.4)	1.00		1.00	
	Monthly	26 (5.4)	1.07 (0.72–1.59)	0.741	1.11 (0.74–1.65)	0.614
	Weekly	27 (5.4)	1.24 (0.84–1.83)	0.274	1.37 (0.92–2.03)	0.123
	Daily	25 (10.5)	1.76 (1.18–2.64)	0.006	1.70 (1.11–2.61)	0.016
			*P*_trend_ = 0.004		*P*_trend_ = 0.008	
Cancer mortality	Seldom	300 (3.9)	1.00		1.00	
	Monthly	20 (4.2)	1.37 (0.87–2.16)	0.175	1.49 (0.94–2.36)	0.087
	Weekly	17 (3.4)	1.24 (0.78–2.02)	0.398	1.30 (0.79–2.13)	0.300
	Daily	8 (3.4)	0.97 (0.48–1.96)	0.929	0.89 (0.43–1.83)	0.747
			*P*_trend_ = 0.889		*P*_trend_ = 0.972	

^1^ Incidence density: number of cases per 1000 person-years. ^2^ Adjusted for age, sex, race, educational levels, occupation, family income, BMI, physical activity, smoking, drinking, disease histories (CVD, cancer, diabetes, hypertension, COPD, hyperlipidemia), and HEI (calculated from 24-dietary interviews).

**Table 3 nutrients-16-03770-t003:** Results of MR drinking on all-cause mortality for the stratified analyses by age, sex, and BMI.

		Seldom	Monthly	Weekly	Daily	
			HR (95%CI) ^1^	HR (95%CI) ^1^	HR (95%CI) ^1^	*P* for Interaction ^2^
Age						0.422
	<=60 years old	1.00	1.69 (1.04–2.75)	1.43 (0.88–2.34)	0.95 (0.47–1.94)	
	>60 years old	1.00	1.18 (0.92–1.52)	1.55 (1.22–1.98)	1.70 (1.29–2.24)	
Sex						0.003
	Female	1.00	0.97 (0.70–1.35)	1.68 (1.26–2.24)	2.01 (1.40–2.90)	
	Male	1.00	1.81 (1.33–2.45)	1.36 (0.97–1.91)	1.24 (0.85–1.81)	
BMI						0.541
	<25 kg/m^2^	1.00	1.62 (1.11–2.37)	1.61 (1.13–2.31)	1.64 (1.12–2.39)	
	>=25 kg/m^2^	1.00	1.08 (0.80–1.46)	1.54 (1.15–2.06)	1.47 (1.00–2.17)	

^1^ Adjusted for age (not in age-stratified models), sex (not in sex-stratified models), race, educational levels, occupation, family income, BMI (not in BMI-stratified models), physical activity, smoking, drinking, disease histories (CVD, cancer, diabetes, hypertension, COPD, hyperlipidemia), and HEI (calculated from 24-dietary interviews). Respondents who seldom drank MR were the reference group. ^2^ *p* values for interaction were calculated by likelihood ratio tests.

## Data Availability

The data that support the findings of this study are publicly available on the National Health and Nutrition Examination Survey (NHANES) website: https://www.cdc.gov/nchs/nhanes/index.htm (accessed on 13 November 2022).The linked mortality data can be found at https://www.cdc.gov/nchs/data-linkage/mortality-public.htm (accessed on 8 May 2023).

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
