# Peer review of "The Frequency of Meal-Replacement Products Drinking and All-Cause, CVD, and Cancer Mortality"

_nutrients, 2024, doi:10.3390/nu16213770_

Round 1

Reviewer 1 Report

Comments and Suggestions for Authors

The manuscript is valuable for the number of the study population followed for a long period of time, 14. 4 years. During this time a total of 1668 deaths appeared, therefore a lot of conclusions can be drawn. However, there are several limitations of this study and they are pointed out correctly in Conclusions, but they tend to diminish the scientific value of the results. This data would be of particular interest in a prospective cohort or a more recent interview inclusion.  

Author Response

Comments1:The manuscript is valuable for the number of the study population followed for a long period of time, 14. 4 years. During this time a total of 1668 deaths appeared, therefore a lot of conclusions can be drawn. However, there are several limitations of this study and they are pointed out correctly in Conclusions, but they tend to diminish the scientific value of the results. This data would be of particular interest in a prospective cohort or a more recent interview inclusion.  

Response1:

We sincerely appreciate your comments on our manuscript. The length of follow-up time is one of the strengths of using NHANES and linked mortality data, which helps us accumulated adequate number of deaths to conduct scientific research. We have pointed out several limitations in the manuscript. One of the main limitations of our study is that we are unable to use more recent cycles in NHANES because the food frequency questionnaires containing the frequency of meal-replacements were not included in NHANES since 2007. Therefore, if future studies are interested in the long-term health effects of meal-replacements, large population-based prospective cohorts were needed in which the meal-repalcements related information should be collected at the recruitment of the study participants.

We have made several refinements to the introduction part and the changes are highlighted, which were shown as follows:

Meal-replacement (MR) products are processed products containing essential nutrients and meager calories that can be substituted for one or two meals in one day[1]. Evidence has proven its effectiveness in weight control, especially for patients with diseases in need of food restriction (such as type 2 diabetes), and that they are safe to use under medical supervision or specialized guidance for patients suffering from severe diseases and being unable to acquire adequate nutrition from normal food[2-5]. 

While many studies have reported their safety, reliability and efficacy in randomized trials[6], there are also concerns regarding the long-term health effects of MR products. MR products are usually rich in protein and low in lipids, which could induce deficiencies in micronutrients[7]. It has also been reported that users drinking MR for weight-loss purposes might suffer substantial weight regain in the long term and subsequently suffer from health problems[8]. Although there are studies reporting the associations of ultra-processed foods and energy-restricted diets[9-11] with elevated risks of mortality, evidence assessing the long-term effects of MR is still insufficient. Moreover, previous studies are mainly based on clinical trials, indicating that the uses of MR products in these studies were usually under strict supervision, and their target populations are usually subjects with obesity or other mild or severe health problems, restricting the generalizability of these studies to general population. Therefore, as the market for MR is expanding and more weight-loss programs are applying these products[12], it is essential and of great importance for public health to investigate the effects of MR on health outcomes in a more general situation.

Reviewer 2 Report

Comments and Suggestions for Authors

Reviewer report

The Frequency of Meal-Replacement Products Drinking and All-cause, CVD and Cancer Mortality

This is a well-organized study showing prospective analysis to assess the associations of meal Replacement Products Drinking and All-cause, CVD, and Cancer Mortality based on the nationally representative dietary and mortality data from the National Health and Nutrition Examination Survey (NHANES). The study shows a well-explained introduction, background, and appropriate results. There are a few sections where the manuscript can be improved.

a. Please describe the entire article in the form of a graphical abstract.

b. How the results are in relation to previously reported studies.

c. This article lacks a diagrammatic representation, describe the mechanism of cardiovascular death by meal replacement products in detail.

d. This study should include a separate paragraph showing the significance of the study outcomes.

e. Did authors mention a Comparison of changes over time for cardiovascular risk factors like triglycerides, high-density lipoprotein, low-density lipoprotein, systolic blood pressure, and diastolic blood pressure?

f. Do the study outcomes remain the same in the case of male and female patients, or do they differ please justify.

Author Response

Comments1:The Frequency of Meal-Replacement Products Drinking and All-cause, CVD and Cancer Mortality

This is a well-organized study showing prospective analysis to assess the associations of meal Replacement Products Drinking and All-cause, CVD, and Cancer Mortality based on the nationally representative dietary and mortality data from the National Health and Nutrition Examination Survey (NHANES). The study shows a well-explained introduction, background, and appropriate results. There are a few sections where the manuscript can be improved.

Response1: 

Thank you so much for the appreciation of our manuscript and comments to improve our manuscript. We have made our efforts to try to address the comments point by point and the responses are listed as follows:

Commens2:Please describe the entire article in the form of a graphical abstract.

Response2:

We have drawn a graphical abstract and sent it to editor Alaina Zhou via email and the graphic abstract is shown below:

Graphic abstract

Comments3: How the results are in relation to previously reported studies.

Response3: 

Previously reported studies focusing on meal-replacement(MR) products are mainly based on randomized trials to assess their efficacy on reducing weight, diet management or refining internal health-related factors, usually ended lasted for one or two years. There is little observational study based on general population to examine their associations with long-term health outcomes. We therefore added some comparisons in the discussion part to show the relationships of our results with previously reported studies(line275-294):

There is little observational study directly assessing the long-term health effects of MR drinking. Previous conclusions about MRs are mostly drawn from randomized trials and they usually put their focus on the efficacy of MR on weight management or their safety for medical use[3, 6, 16, 19-21]. One systematic have summarized that incorporating MR into weight loss programmes would benefit the users from reducing weight and cardiometabolic risks within a relative short time of follow-up(longest follow-up time included: ≤52 weeks). However, another systematic review pointed out that the conclusions become inconsistent when it come to long-term weight loss(>1 year). Besides, the follow-up time of these trials were usually inadequate to accumulate enough disease outcomes or mortality. The food frequency data from NHANES 2003-2006 and the linked mortality data allowed us to have sufficient follow-up time (median: 14.4 years) to evaluate the associations between MR and mortality during a more extended period and suggest its potential long-term health effects in a more general population. In addition, in our mlutivariate-adjusted models, we additionally adjusted for the HEI of each respondent and showed that the effects of MR on mortality were independent of dietary quality. However, it should be acknowledged that only a limited number of respondents were included in our analysis. We did try to maximize our sample size using multiple imputation in the sensitivity analysis though, the prospective results should be validated in larger samples and more studies.

Comments4: This article lacks a diagrammatic representation, describe the mechanism of cardiovascular death by meal replacement products in detail.

Response4: 

After a literature reviewing, we found there was almost no study that explicitly explored the mechanisms of MRs on cardiovascular death. We are unable to draw a diagrammatic representation due to the lack of sufficient evidence. However, MRs are one kind of high protein products, and several observational studies have proposed that long-term high-protein intake might contribute to the declined renal function, which are associated with increased risks of cardiovascular diseases. We assume that the ‘MRs - declined renal function - cardiovascular death’ chain would be one of the underlying mechanisms of cardiovascular death by MRs. We have added additional explanations about the potential mechanisms of cardiovascular death caused by MRs in Line 314-319 as:

As one kind of high protein products, several observational studies have proposed that long-term high-protein intake might contribute to the decline of renal function and finally to increasing the risk of chronic kidney disease(CKD) in individuals with or without preexisting CKD due to intra-glomerular pressure caused by high-protein intake[28]. The changes of renal function might perhaps subsequently lead to increased risks of cardiovascular diseases and diverse health conditions[29,30].

Comments5:This study should include a separate paragraph showing the significance of the study outcomes.

Response5: 

We have added one separate paragraph showing the significance of the study outcomes as follows(Line:334-342):

Our study may have important implications for public health, as well as future investigations into this topic. First of all, previous studies have mostly reported benefits of MR use, but our study pointed out that frequent use of MRs could increase the risk of all-cause mortality. Therefore, our study may raise concerns of MR users about the potential long-term health problems caused by frequently drinking of MRs, especially for daily drinkers. For future studies, our study indicated that an extended follow-up time would be needed to fully assess the impacts of MRs on health, and our results might be of interest to studies including cohorts who want to conduct nutrition- or dietary behaviors-related surveys.

Comments6: Did authors mention a Comparison of changes over time for cardiovascular risk factors like triglycerides, high-density lipoprotein, low-density lipoprotein, systolic blood pressure, and diastolic blood pressure?

Response6: In the main manuscript, we didn’t mention the comparison of changes over time for cardiovascular risk factors. In addition, we were to unable capture these changes because NAHENS is a set of cross-sectional studies with different respondents visited each cycle. Besides, the longitudinal comparison of cardiovascular risk factors cannot be related to MR drinking because the the food frequency questionnaires containing the frequency of meal-replacements were not included in NHANES since 2007.

Comments7: Do the study outcomes remain the same in the case of male and female patients, or do they differ please justify.

Response7:

Thank you for the comment. In the stratified analyses, we did observe interaction of MR drinking with sex on all-cause mortality, and the study results were different in the case of males and females. We have made this results more clearly shown in the main text as(Line 200-204): In stratified analyses, we found significant interactions of MR drinking with sex on all-cause mortality (Table 3, P for interaction: 0.003). The effects were stronger among females than males for weekly and daily drinkers. The aHRs and 95%CI for weekly and daily female drinkers were: 1.68 (1.26-2.24) and 2.01 (1.40-2.90), respectively. The aHRs and 95%CI for weekly and daily male drinkers were: 1.36 (0.97-1.91) and 1.24 (0.85-1.81), respectively.

Reviewer 3 Report

Comments and Suggestions for Authors

The authors examined the relationship between the consumption of meal-replacement products and total mortality, CVD mortality, and cancer mortality using the NHANES dataset. The use of these foods is expected to increase, thus this study may be meaningful, but the reviewer has objections about the purpose of using these foods. Authors describe these products have “effectiveness in weight control, especially for patients with diseases 34 in need of food restriction (such as type 2 diabetes)” (line 34-35). However, these products are also used among those who, because of some reason, are unable to obtain sufficient nutrition from a normal diet. For example, people who are physically disabled and unable to prepare their own meals, people with chewing problems due to missing teeth or paralysis of the oral muscles, people who cannot consume enough energy due to a consumptive disease such as COPD, cancer.

Authors should have carefully considered the reason the participants had the MR drinks, either to supplement the nutritional deficiencies (COPD, heart failure, cancer patients, etc.) or to restrict the nutrients the person is taking (diabetes patients, etc.). The prognosis for those with anorexia or malnutrition caused by these diseases is generally poor. These concerns of the reviewer are actually expressed in Supplementary Table 2. Exclusion those with past history of CVD, cancer, and death within first 2 years resulted in the disappearance of significant difference in HRs.

In order to verify the authors' claim that “MR foods are used for weight loss”, the reviewer thinks that the analysis should have been limited to “people who need/want to lose weight”. If authors can use data such as “weight loss intentions” or serum albumin levels, these will be useful for these considerations.

L112 “we additionally calculated the Healthy Eating Index (HEI)”

Authors should describe whether the HEI calculation was based on the results of the 24-hour recall or the FFQ.

Table 2.

Please indicate whether the nutritional intake data in Table 1 is based on the results of a 24-hour recall or a FFQ.

Author Response

Comments1: 

The authors examined the relationship between the consumption of meal-replacement products and total mortality, CVD mortality, and cancer mortality using the NHANES dataset. The use of these foods is expected to increase, thus this study may be meaningful, but the reviewer has objections about the purpose of using these foods. Authors describe these products have “effectiveness in weight control, especially for patients with diseases 34 in need of food restriction (such as type 2 diabetes)” (line 34-35). However, these products are also used among those who, because of some reason, are unable to obtain sufficient nutrition from a normal diet. For example, people who are physically disabled and unable to prepare their own meals, people with chewing problems due to missing teeth or paralysis of the oral muscles, people who cannot consume enough energy due to a consumptive disease such as COPD, cancer.

Authors should have carefully considered the reason the participants had the MR drinks, either to supplement the nutritional deficiencies (COPD, heart failure, cancer patients, etc.) or to restrict the nutrients the person is taking (diabetes patients, etc.). The prognosis for those with anorexia or malnutrition caused by these diseases is generally poor. These concerns of the reviewer are actually expressed in Supplementary Table 2. Exclusion those with past history of CVD, cancer, and death within first 2 years resulted in the disappearance of significant difference in HRs.

In order to verify the authors' claim that “MR foods are used for weight loss”, the reviewer thinks that the analysis should have been limited to “people who need/want to lose weight”. If authors can use data such as “weight loss intentions” or serum albumin levels, these will be useful for these considerations.

Response1: 

Thank you so much for the suggestions regarding the descriptions of purposes for drinking MR. We totally agree that we should be more cautious about the purposes for using MR because disabled respondents who are unable to eat by themselves or respondents with severe diseases are also in need of energy supplementation besides respondents with obesity or demands for weight control. Therefore, we made some changes to the description of purposes for using MR and the changes are highlighted. However, in NHANES most of the respondents were capable of finishing questionnaires and physical examinations. Only 8 (less than 1 %)respondents had much difficulty in eating by themselves in our study population(6,770 respondents) and there were 421 respondents(6.2%) who reported being unable to eat because of problems with teeth, mouth or dentures. Besides, after excluding those who reported being unable to eat because of problems with teeth, mouth or dentures, the results remained consistent with the main results and were shown below and in the sensitivity analysis(Supplementary Table 2):

We have also tried some refinements on our analyses to make our results more robust. First, we tried to use the variable ‘Weight loss/low cal/low carb/hi pro diet’ to distinguish respondents with weight loss intentions. However, the number of respondents selected through this variable is too few to provide sufficient statistical power(only hundreds). Next, we used the variable ‘Tried to lose weight in the past year’ and ‘Tried not to gain weight in the past year’ to keep respondents who had weight loss or weight retain intentions(N=3,082). The associations of MR drinking with mortality in respondents who had weight loss or weight retain intentions were shown in sensitivity analysis(Supplementary Table 2). The associations of weekly MR drinking with all-cause mortality remained significant. We assumed that the disappearance of significance in the associations of daily MR drinking with all-cause mortality were due to reduced sample size (only 69 respondents drank MR daily after exclusion), and we therefore combined the ‘weekly’ and ‘daily’ group again to increase the sample size and the results were shown below: The associations of combined MR  drinking(weekly and daily) with all-cause mortality were significant.

additional results1

In the sensitivity analysis of excluding baseline cases with diabetes, we further excluded respondents who is on diabetic diet or other special diet to exclude potential respondents who drink MR products for special reasons including nutrition restriction. Third, as suggested, since the serum albumin levels are related to internal nutrition levels, we excluded respondents with abnormal low levels of serum albumin to exclude potential respondents who are likely to be affected by severe diseases and be unable to acquire adequate nutrition. The added results are shown in the supplementary materials(Supplementary Table 2) and table below and are consistent with the main results for all-cause mortality:

additional results2

Comments2: L112 “we additionally calculated the Healthy Eating Index (HEI)”

Authors should describe whether the HEI calculation was based on the results of the 24-hour recall or the FFQ.

Response2: 

The HEI scores were all calculated from 24-hour recall, and we have added additional declaration in the notes of the tables and main text.

Comments3: 

Table 2.

Please indicate whether the nutritional intake data in Table 1 is based on the results of a 24-hour recall or a FFQ.

The nutritional intake were all calculated from 24-hour recall, and we have added additional declaration in the notes of the tables.

Round 2

Reviewer 3 Report

Comments and Suggestions for Authors

The manuscrip was appropriately revised.

Just one thing, "Serum Albumin"(L125-126) should be "Serum albumin".